# Co-occurrence of ST412 *Klebsiella pneumoniae* isolates with hypermucoviscous and non-mucoviscous phenotypes in a short-term hospitalized patient

Qinghua Liang,[1,2] Nan Chen,[1] Wei Wang,[1] Biying Zhang,[1] Jinjing Luo,[1] Ying Zhong,[1] Feiyang Zhang,[1] Zhikun Zhang,[1] Alberto J. Martín–Rodríguez,[3,4] Ying Wang,[1] Li Xiang,[1] Xia Xiong,[5] Renjing Hu,[6] Yingshun Zhou[1,7]

**ABSTRACT**   Hypermucoviscosity (HMV) is a phenotype that is commonly associated with hypervirulence in *Klebsiella pneumoniae*. The factors that contribute to the emergence of HMV subpopulations remain unclear. In this study, eight *K. pneumoniae* strains were recovered from an inpatient who had been hospitalized for 20 days. Three of the isolates exhibited a non-HMV phenotype, which was concomitant with higher biofilm formation than the other five HMV isolates. All eight isolates were highly susceptible to serum killing, albeit HMV strains were remarkably more infective than non-HMV counterparts in a mouse model of infection. Whole genome sequencing (WGS) showed that the eight isolates belonged to the K57-ST412 lineage. Average nucleotide identity (FastANIb) analysis indicated that eight isolates share 99.96% to 99.99% similarity and were confirmed to be the same clone. Through comparative genomics analysis, 12 non-synonymous mutations were found among these isolates, eight of which in the non-HMV variants, including *rmpA* (c.285delG) and *wbaP* (c.1305T > A), which are assumed to be associated with the non-HMV phenotype. Mutations in *manB* (c.1318G > A), *dmsB* (c.577C > T) and *tkt* (c.1928C > A) occurred in HMV isolates only. RNA-Seq revealed transcripts of genes involved in energy metabolism, carbohydrate metabolism and membrane transport, including *cysP*, *cydA*, *narK*, *tktA*, *pduQ*, *aceB*, *metN,* and *lsrA*, to be significantly dysregulated in the non-HMV strains, suggesting a contribution to HMV phenotype development. This study suggests that co-occurrence of HMV and non-HMV phenotypes in the same clonal population may be mediated by mutational mechanisms as well as by certain genes involved in membrane transport and central metabolism.

**IMPORTANCE**   *K. pneumoniae* with a hypermucoviscosity (HMV) phenotype is a community-acquired pathogen that is associated with increased invasiveness and pathogenicity, and underlying diseases are the most common comorbid risk factors inducing metastatic complications. HMV was earlier attributed to the overproduction of capsular polysaccharide, and more data point to the possibility of several causes contributing to this bacterial phenotype. Here, we describe a unique event in which the same clonal population showed both HMV and non-HMV characteristics. Studies have demonstrated that this process is influenced by mutational processes and genes related to transport and central metabolism. These findings provide fresh insight into the mechanisms behind co-occurrence of HMV and non-HMV phenotypes in monoclonal populations as well as potentially being critical in developing strategies to control the further spread of HMV *K. pneumoniae*.

**KEYWORDS**   *Klebsiella pneumoniae*, RNA-Seq, hypermucoviscosity, non-synonymous mutations

Address correspondence to Yingshun Zhou, yingshunzhou@swmu.edu.cn, or Renjing Hu, weiweihuhu112@163.com.

Qinghua Liang and Nan Chen contributed equally to this article. Author order was determined based on their contribution to the article.

The authors declare no conflict of interest.

See the funding table on p. 14.

*K*lebsiella pneumoniae exhibiting phenotypic hypermucoviscosity (HMV) is frequently associated with hypervirulence. The emergence of HMV, hypervirulent and multidrug resistant isolates is raising global concerns and a gradual increase in the morbidity and mortality of *K. pneumoniae* infections (1), which include urinary tract infections, meningitis, pyogenic liver abscesses, and empyema, among others (2).

HMV is a prominent phenotypic feature characterized by the formation of a viscous filament >5 mm when the colony is stretched by an inoculation loop on an agar plate (3). The correlations between the HMV phenotype and certain specific regulatory factors are widely accepted. For instance, the mucoid regulator gene, *rmpA,* located on a large virulence plasmid, has been shown to be required for HMV phenotype development and as such, deletion of *rmpA* resulted in reduced amount of capsule and a non-HMV phenotype (4). Moreover, *rmpC* and *rmpD*, neighboring *rmpA*, are also involved in the hypermucoviscosity of hypervirulent *K. pneumoniae* (Hvkp). Notably, genetic studies showed that *rmpD* deletion resulted in an HMV-negative phenotype and did not down-regulate capsular polysaccharide (CPS) content, whereas *rmpC* mutation reduced CPS biosynthesis but retained the HMV phenotype, thus highlighting that capsule synthesis is not the only factor required for the expression of HMV (5, 6). HMV has also been attributed to CPS overproduction, albeit it does not fully account for the presence of an HMV phenotype (7). Most genes in the *cps* cluster, including *wza, wzc, wzy,* and *wckU,* are involved in the transport and assembly of CPS (8). For example, RmpD binds Wzc, thereby altering Wzc-mediated CPS synthesis to produce longer and more uniformly long polysaccharide chains, which is likely an essential phenotypic factor in HMV (9). A non-HMV phenotype caused by mutations in the *cps* cluster has been shown to occur among clinical carbapenem-resistant *K. pneumoniae* isolates, affecting susceptibility to carbapenems but also resulting in a better ability to form biofilm (10).

A number of genes associated with energy homeostasis seem to be important for HMV phenotype (11). A recent analysis of a library of transposon integration mutants of the hypervirulent strain KPPR1 also suggested that several disrupted metabolism-related genes have diverse effects on HMV and CPS production (6). Thus, six genes induced HMV loss and CPS reduction (*galU, rfaH, wzyE, arnD, arnE,* and *wcaJ*), five resulted in HMV loss but did not affect CPS content (*uvrY, miaA, galF, arnF,* and *orf2*) and two reduced CPS content but did not affect HMV(*rnfC2* and *arcB*) (6). These data suggest that metabolism-related genes are associated with the synthesis of HMV. Therefore, mutations or alterations in the expression of these genes or others with related functions might contribute to the emergence of HMV. In addition to metabolism, other factors influence the HMV phenotype, such as antibiotic stress or the host immune system, suggesting that the factors influencing HMV are diverse (12). The full breadth of determinants behind HMV formation is yet unknown and their investigation is essential for an improved prevention and control of *K. pneumoniae* infections.

In this study, eight ST412-K57 *K. pneumoniae* isolates derived from the same clonal population were obtained from a short-term hospitalized patient and the HMV phenotypes were investigated. Whole genome sequencing (WGS), RNA-Seq, CPS quantification, serum killing assays, biofilm formation tests, siderophores secretion assays, and virulence tests were performed to elucidate the factors contributing to HMV evolution of *K. pneumoniae in vivo* in patients, as well as the selective advantage of HMV phenotypic variation in terms of biofilm production and siderophore biosynthesis.

## RESULTS

### CPS content is associated with the HMV phenotype

Eight bacterial strains were found in the blood, urine, feces, and cerebrospinal fluid of a patient with underlying medical disorders (diabetes mellitus and hypertension) (Fig. 1). The VITEK two compact System (bioMérieux, France) identified the eight isolates as *K. pneumoniae*. Antimicrobial susceptibility testing showed that the eight *K. pneumoniae* isolates were susceptible to all tested antibiotics (Table S1). We noticed HMV phenotypic variation in eight isolates. To investigate the HMV phenotype of the eight *K. pneumoniae*

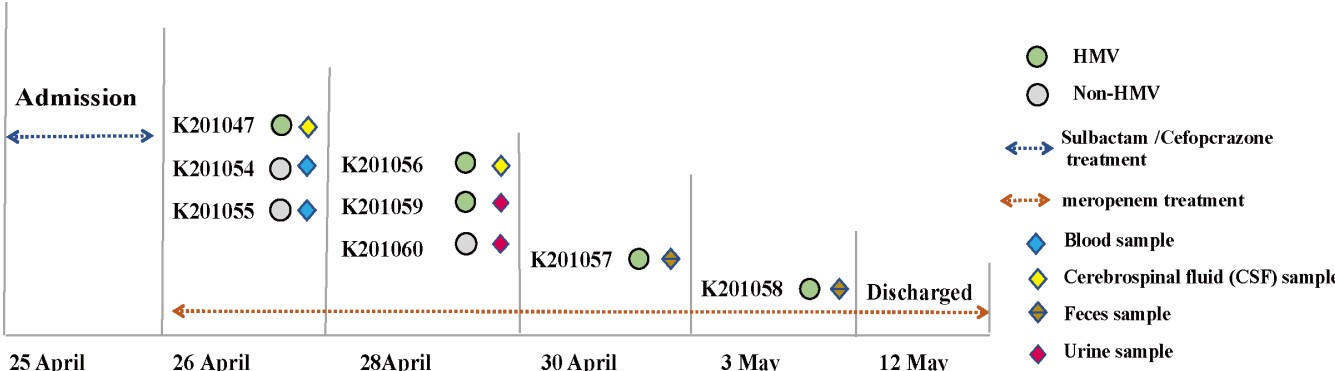

**FIG 1** Timeline of the eight episodes of *K. pneumoniae* infections in the inpatient. The diamonds of various colors represent the specimen source. Grey circles represent the non-HMV phenotype, and green circles indicate the HMV phenotype. The colored dotted line represents the antibiotic used for treatment.

isolates retrieved during the infection episode, we first performed the string test on colonies grown on agar plates. This test revealed that five isolates were HMV (K201047, K201056, K201057, K201058, and K201059), whereas the other three were non-HMV (K201054, K201055, and K201060) (Fig. 2A). HMV is associated with poor sedimentation. Therefore, to validate the string test observations, we performed a natural sedimentation assay in which bacteria were allowed to sediment by gravity. The supernatant of HMV strains remained turbid with a mean $OD_{600}$-supernatant/$OD_{600}$-total ratio of approximately 0.5, whereas the mean value of non-HMV isolates did not exceed 0.25 (Fig. 2B). In addition, given that the majority of HMV strains have the ability to overproduce capsule, we next extracted the CPS. Consistently, the capsule content of the HMV isolates was significantly higher compared to that of non-HMV isolates ($P < 0.05$) (Fig. 2C). Taken together, our results confirmed that there is a correlation between differences in sedimentation and the level of capsular material produced by the isolates.

## Non-HMV strains show increased biofilm formation and reduced pathogenicity

To investigate the association between phenotypic HMV and other properties relevant to *K. pneumoniae* infectivity and persistence, we first evaluated whether there was a correlation between the different mucoid phenotypes with biofilm formation on the isolates. It was found that the non-HMV clinical isolates had 3-fold higher biofilm-forming capacity than HMV isolates, as determined by crystal violet staining of total biofilm biomass (Fig. 3A). Production of siderophores such as yersiniabactin, aerobactin, or salmochelin, are known to contribute to *K. pneuominiae* virulence (13). To investigate differences in siderophore production between HMV and non-HMV strains, we performed a Chrome Azurol S (CAS) agar assay, in which siderophore production is phenotypically determined by the generation of an orange halo (Fig. S1). This test revealed that the siderophore secretion capacity of non-HMV isolates was higher than that of HMV isolates; however, the difference was not statistically significant ($P = 0.0663$) (Fig. 3B). Next, we tested whether the HMV phenotype offered protection against serum-mediated killing. As shown in Fig. 3C, although the serum resistance of K201055 and K201060 in the non-HMV group was enhanced at 1 h, all isolates were completely eradicated after three hours of co-incubation with human serum, clearly indicating that all isolates were highly susceptible to serum killing. To evaluate the pathogenicity of HMV isolates and non-HMV isolates *in vivo*, a mouse model of intraperitoneal infection was employed. The bacterial loads in the liver, spleen, and kidney of infected mice were quantified following intraperitoneal inoculation of $5.5 \times 10^7$ CFUs throughout a 24 h infection. The results showed that the non-HMV isolates had colonization levels about three logs lower than HMV isolates (Fig. 3D through F), suggesting that virulence of HMV *K. pneumoniae* isolates is increased.

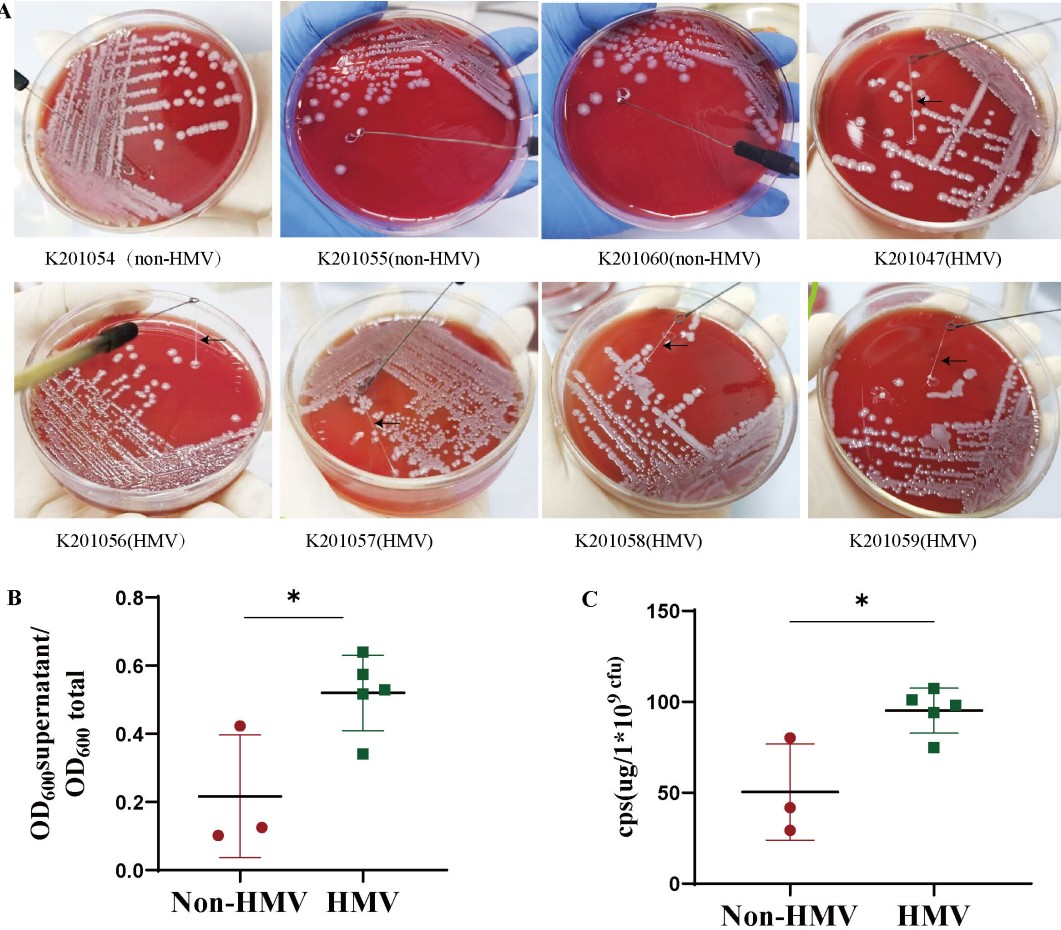

**FIG 2** The mucoviscosity phenotype and capsule production of the eight clinical isolates. Each assay was carried out in triple with three independent samples. (A) Colony morphology of the isolates cultured on blood agar plates. (B) Sedimentation properties of the isolates. (C) Amounts of CPS produced as determined by the uronic acid assay. Each assay was independently repeated at least three times with three independent samples. The data represent the mean ± SD, and asterisks indicate significant differences (*$P < 0.05$) between the compared groups using $t$-test.

## WGS revealed that the eight isolates belonged to the same clone

To investigate the genomic characteristics of the co-occurrent HMV and non-HMV *K. pneumoniae* isolates, we performed whole genome sequencing (WGS) on the eight strains recovered. Analysis of genomic data revealed that all strains belonged to the K57-ST412 lineage, with an average genome size of 5.6 Mb, a GC content of approximately 57.4%, and BLAST-based Average Nucleotide Identity (ANIb) values ranging from 99.96% to 99.99% between isolates. These results confirmed the intimate genetic relatedness of the eight isolates. Pulsed-field gel electrophoresis (PFGE) profiling divided the eight isolates into two main subtypes, one composed of the non-HMV isolates and strain K201059, and the second formed by the remaining HMV strains (Fig. S2). Additionally, two plasmids, designated pA and pB, were found in all eight isolates after plasmid content analysis using PlasmidFinder (14). Plasmid pA contained an IncFIB(k) replicon, and exhibited a high similarity with pGN-2, a plasmid harbored by the *K. pneumoniae* strain GN-2 (99.86% identity and 68% coverage, accession no. NZ_CP019161.1), albeit with a higher number of transposases and hypothetical protein coding genes. Plasmid pB was identical in sequence to pBio19 from *K. pneumoniae* strain Bio19, isolated in Turkey and Iraq (99.98% identity and 100% coverage, accession no. NZ_CP096811.1) (Fig. S3). BLAST analysis against the virulence genes database revealed the isolates contained several virulence determinants located on plasmid A, including genes that determine

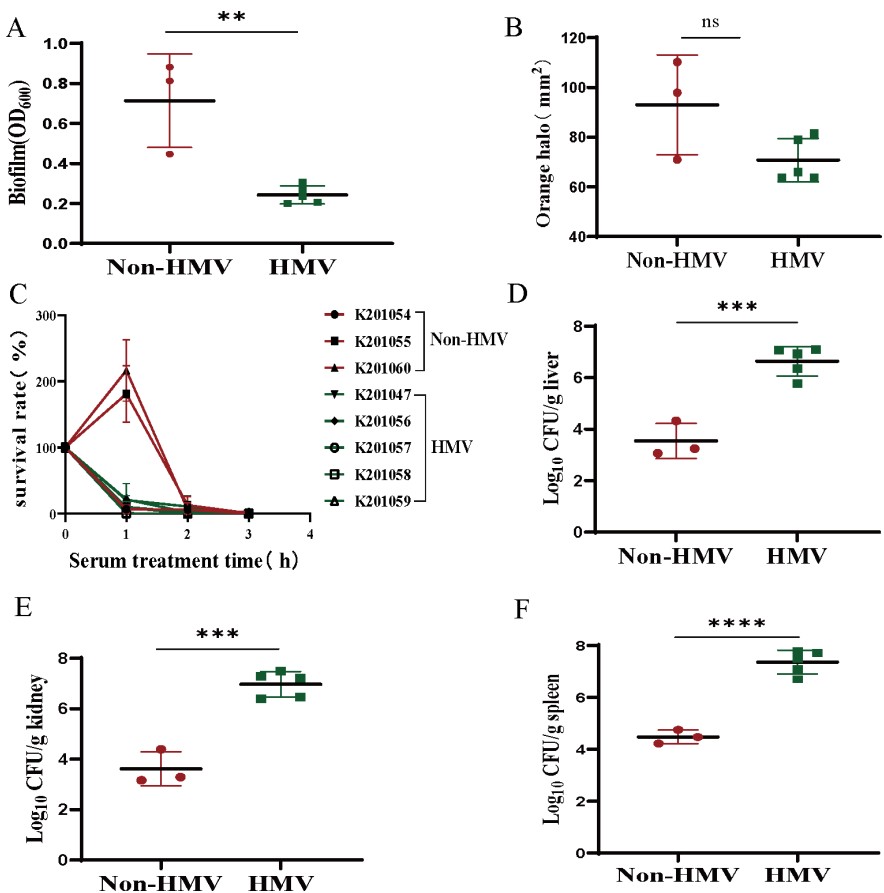

**FIG 3** Phenotype and infectivity of eight *K. pneumoniae* isolates. (A) Crystal violet assay for assessment of biofilm formation of eight isolates. (B) Siderophore production assessed by the CAS assay. (C) Rate of HMV and non-HMV isolates surviving in human serum. (D) Bacterial loads of HMV and Non-HMV isolates in liver tissues. (E) Bacterial loads of HMV and Non-HMV isolates in kidney tissues. (F) Bacterial loads of HMV and Non-HMV isolates in spleen tissues. Bacterial numbers (expressed as $\log_{10}$ CFU) were standardized per 0.1 g of wet organ weight. All assays were carried out in triplicate with three independent samples. The data represent the mean ± SD, $P$ values were calculated using the $t$ test, and asterisks indicate significant differences: $P > 0.05$ (ns), $P < 0.01$ (**), $P < 0.001$ (***), and $P < 0.0001$ (****).

major mucoid phenotypes (*rmpA*, *rmpC,* and *rmpD*), genes for the biosynthesis or uptake of iron (*iutA*, *iroBCDN*, *irp2* and *ybts*).

## A single-nucleotide deletion in *rmpA* and a base substitution in *wbaP* might be responsible for the non-HMV phenotype

To gain an insight on the genomic determinants behind HMV development in co-occurring *K. pneumoniae* isolates, we used the genome sequence of K201047 as a reference and performed a whole-genome alignment of all other seven isolates against the K201047 genome. Our analysis identified 193 mutation events (SNPs and InDels), most of which were either located in intergenic regions (119 mutations) or were synonymous mutations (62 mutations), with only 12 mutations being non-synonymous. Subsequently, to explore the genetic determinants associated with the HMV phenotypic changes, we first focused on the non-synonymous mutations that occurred in the non-HMV strains and not in the HMV strains (Table 1). Firstly, we identified a G-base deletion that disrupts the *rmpA* gene(c.285delG), a regulator of mucoid phenotype, which was predicted to be responsible for the non-HMV phenotype in strain K201054. In the other two non-HMV isolates, K201055 and K201060, a single missense mutation was found in the *wbaP* gene (c.1305T > A), which encodes a undecaprenyl-phosphate galactose phosphortransferase

**TABLE 1** Non-synonymous mutation and InDels found in the strains

| Gene | K201047 | K201054 | K201055 | K201056 | K201057 | K201058 | K201059 | K201060 | Product | Nucleotide changes | Protein changes |
|------|---------|---------|---------|---------|---------|---------|---------|---------|---------|--------------------|-----------------|
| *ydcJ* | T | G | T | T | T | T | T | T | Metalloenzyme | c.1133T > G | V378G |
| *rmpA* | T | G(del) | T | T | T | T | T | T | HMV phenotype regulator | c.285delG | R96G |
| *wbap* | T | T | A | T | T | T | T | A | Undecaprenyl-phosphate galactose phosphotransferase | c.1305T > A | S435R |
| *bglA* | A | A | G | A | G | A | A | A | 6-Phospho-beta-glucosidase | c.259A > G | T87A |
| *recC* | G | G | A | G | G | G | G | G | Exonuclease V | c.2538G > A | R846H |
| *minE* | C | C | C | C | C | C | C | T | Cell division topological specificity factor | c.128C > T | A43H |
| *hrpB* | C | C | C | C | C | C | C | A | RNA-dependent NTPase | c.178C > A | V60I |
| *manB* | G | G | G | G | G | A | G | G | Phosphomannomutase | c.1318G > A | G440L |
| *dmsB* | C | C | C | C | C | C | T | C | Dimethylsulfoxide reductase | c.577C > T | R193W |
| *tkt* | C | C | C | C | C | C | A | C | Transketolase | c.1928C > A | A643N |

involved in CPS synthesis. To confirm that mutations in *rmpA* and *wbaP* were involved in the development of a non-HMV phenotype, we constructed complementation plasmids containing the CDS region and its native promoter. Introduction of native *rmpA* into non-HMV strain K201054 and *wbaP* into non-HMV strains K201055 and K201060 restored their HMV phenotypes. (Fig. 4A through C). Besides, we also observed multiple independent mutations in the genes *recC*, *minE*, *ydcJ* and *hrpB*, the latter encoding an ATP-dependent helicase that is known to promote pathogenicity (15). To the best of our knowledge, these four genes have not been documented to be associated with the HMV phenotype, and additional research is needed to investigate their potential contribution.

For the clinical HMV isolates K201057 and K201059, firstly, two distinct mutations related to carbohydrate metabolic pathways were found, one in the *tkt* gene linked to glycolysis, and another in the *bglA* gene involved in the phosphorylated disaccharide metabolism pathway. Additionally, we also observed one independent mutation in the gene *dmsB* in the HMV isolate K201059, encoding the electron-transfer subunit of the dimethylsulfoxide reductase (16). Subsequently, we noticed a base substitution in *manB* in the HMV strain K201058. *manB* is distributed at the *cps* locus and encodes the phosphomannomutase converting enzyme, which catalyzes the conversion between mannose-6-phosphate and mannose-1-phosphate, the latter of which may be involved in CPS assembly (17). It is well known that HMV and the *cps* locus involved in capsular

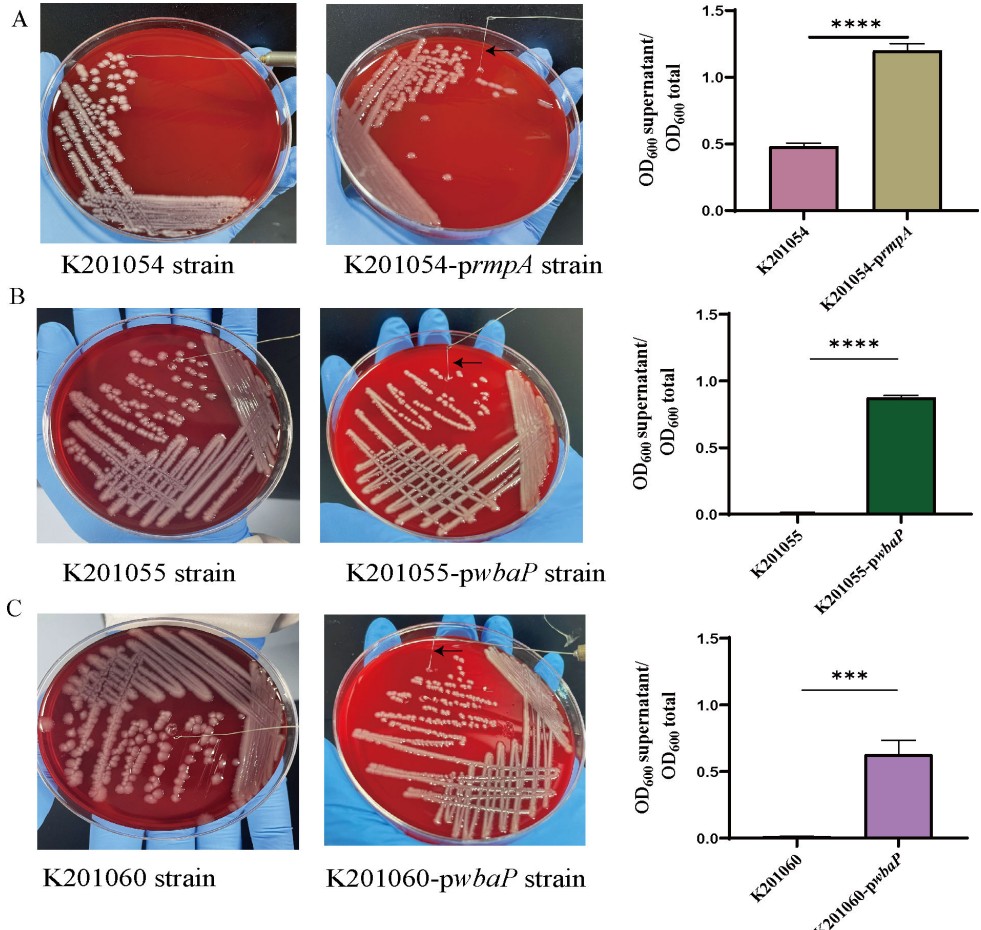

**FIG 4** Genetic complementation of mutants. (A) Complementation of the native *rmpA* gene into the K201054 non-HMV strain restored the HMV phenotype. (B) Complementation of the native *wbaP* gene into the K201055 non-HMV strain restored the HMV phenotype. (C) Complementation of the native *wbaP* gene into the K201060 non-HMV strain restored the HMV phenotype. Natural sedimentation assay was repeated three times. The data represent the mean ± SD, and asterisks indicate significant differences [$P < 0.0001$(****), $P < 0.01$ (***)] between the compared groups using *t*-test.

production are functionally associated (18). In our case, the HMV phenotype of strain K201058 did not seem to be impacted by the *manB* mutation.

## Differentially expressed genes involved in transport and metabolic activities are associated with non-HMV phenotypes

We performed a genome-wide transcriptomic analysis by RNA-seq on eight isolates to identify differences in their transcriptomic profiles that might contribute to their distinct phenotypic characteristics. A total of 100 differentially expressed genes (DEGs) were identified in non-HMV strains compared to HMV strains (|log2 (fold change) |>1, *P* < 0.05), with 39 up-regulated transcripts (39%) and 61 down-regulated transcripts (61%), as shown in the volcano plot (Fig. 5A). The most significantly up-regulated gene was found to be *fimI*, encoding a fimbrial protein subunit, and the most significantly down-regulated gene was *cysP*, encoding the thiosulfate/sulfate ABC transporter periplasmic binding protein, with a log2FC of 2.51 and −2.54, respectively.

A functional annotation analysis was conducted to identify dysregulated gene functions and biological pathways. The differentially expressed genes were mapped to Gene Ontology (GO) categories and KEGG pathways (Table S2). We found that the DEGs were clustered within 10 GO functional categories, with "integral component of membrane" and "plasma membrane" dominating the GO categories (Fig. 5B). Subsequently, a KEGG functional annotation analysis was further performed to categorize DEGs into diverse pathways. As shown in Fig. 5C, the results indicated that "energy metabolism", "carbohydrate metabolism," and "membrane transport", were the most enriched pathways in DEGs, and other major pathways included "infections disease: bacterial", "translation," and "amino acid metabolism".

Based on GO and KEGG analysis, it was found that 8 DEGs associated with carbohydrate metabolism were downregulated. In particular, *pduQ* involved in the propanediol utilization activity, *tktA* encoding the transketolase and *aceB* encoding the malate synthase A were significantly down-regulated in non-HMV isolates (|log2 (fold change) |>1.3). Similarly, *puuA* (glutamine synthetase) and *puuD* (gamma-glutamyl-GABA hydrolase) were also significantly downregulated in amino acid metabolism in non-HMV isolates (|log2 (fold change) |>1.3). Genes related to energy metabolism were also found to be differentially expressed, including a nitrate transporter (*narK*), the methylenetetrahydrofolate reductase (*metF*), the thiosulfate ABC transporter permease (*cysU*), and the cytochrome ubiquinol oxidase subunit I (*cydA*). Interestingly, RNA-Seq data showed that genes participating in lipid metabolism (*glpK*, *fadD*), metabolism of cofactors and vitamins (*iscS*), membrane transport (*lsrA*, *metN*, *lsrC*, and *ftiY*), drug resistance: antimicrobial (*ycfS*, *degP*), were also significantly dysregulated (Fig. 5D). Sixteen DEGs were selected to validate the RNA-Seq data via RT-qPCR analyses, which highly correlated with the RNA-Seq data (Fig. 5E).

## DISCUSSION

As a preponderant bacterium driving community-acquired infections, *K. pneumoniae* with HMV has the ability to cause complicated infections, including pneumonia, liver abscesses, and sepsis (19). Infectious *K. pneumoniae* strains can undergo HMV phenotypic variation in response to diverse hostile environments, with the result that both HMV and non-HMV phenotypes can be observed within the same clonal population. Published data for *K. pneumoniae* ST11 strains demonstrated that switching of the HMV phenotype can provide a fitness advantage for invasive infections (20). In this study, we observed that HMV and non-HMV phenotypes co-occurred in the same clonal isolates. We investigated the underlying mechanisms using whole genome sequencing and RNA-Seq profiling. We speculate that the *rmpA* (c.285delG) mutation not only results in better sedimentation but is also responsible for the non-HMV phenotype of strain K201054, in agreement with evidence associating *rmpA* with the HMV phenotype of *K. pneumoniae* has been confirmed (21, 22). Furthermore, the mutation in *wbaP* (c.1305T > A) resulted in better natural sedimentation and reduced the overproduction of capsule

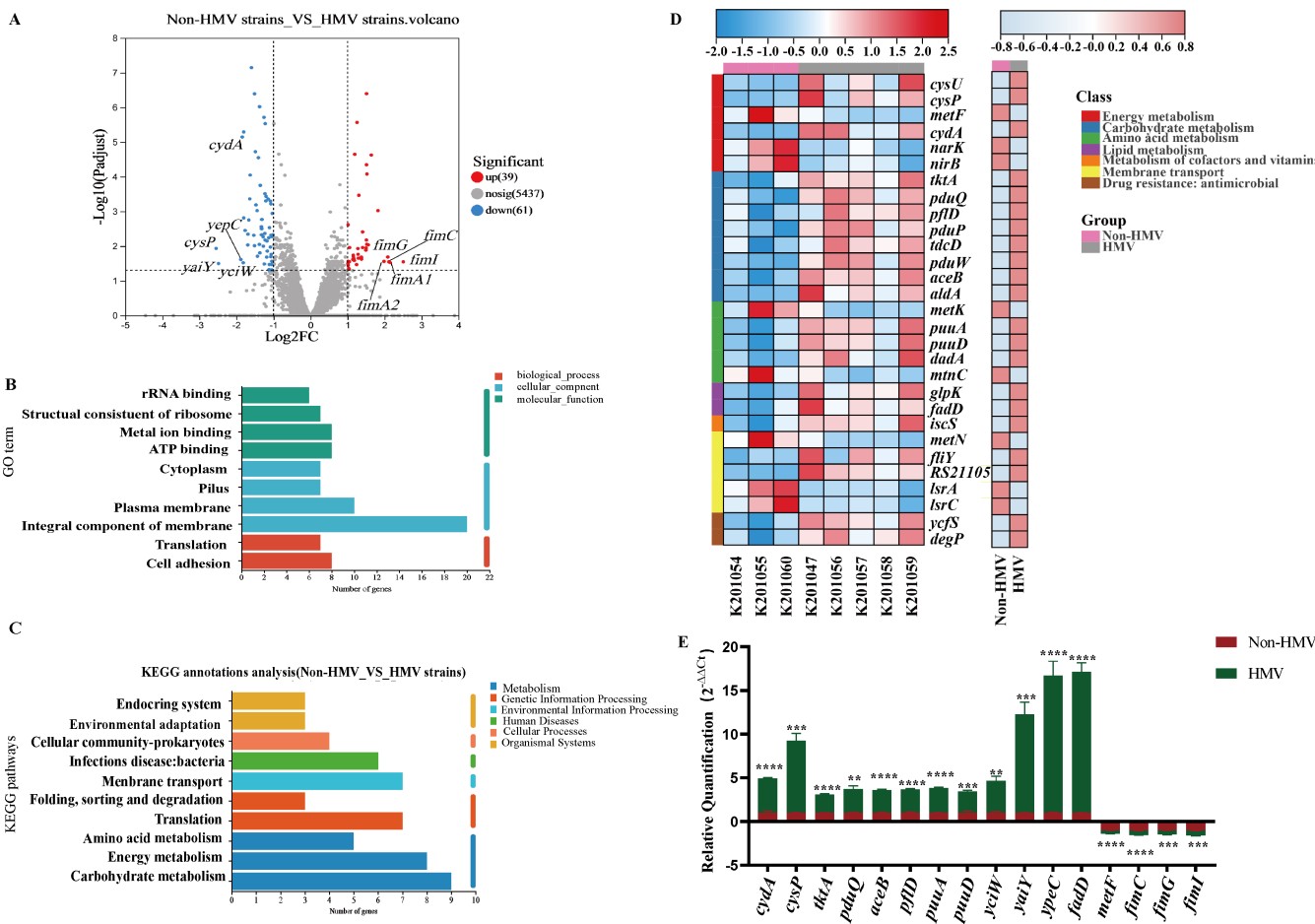

**FIG 5** RNA-seq gene expression profile of non-HMV strains (K201054, K201055, and K201060) compared to that of HMV strains (K201047, K201056, K201057, K201058, and K201059). (A) Volcano plot of DEGs in bacterial cells. -Log10 (P$_{adjust}$) indicates the statistical test value for the difference in gene expression change, that is, the *P* value. (B) GO function annotation analysis of DEGs. Rich factor refers to the number of genes annotated to the GO term in a gene set. (C) Clusters of KEGG pathways. (D) Heat map of typical DEGs associated with energy metabolism, carbohydrate metabolism and amino metabolism in eight strains (non-HMV strains_vs_HMV strains). (E) RNA-Seq data validation by RT-qPCR.

polysaccharide in the non-HMV strains K201055 and K201060. A strong link between capsule production and the HMV phenotype has been reported, despite a few reports suggesting that HMV does not require overproduction of capsule (18). We confirmed a reduction in CPS synthesis in clinical isolates K201055 and K201060, corresponding to a non-HMV appearance on the plates.

Absence of *rmpA* has been associated with higher biofilm formation capacity (23); similarly, mutations in *wbaP* can lead to increased biofilm formation ability (24). Therefore, we hypothesize that the mutations in *rmpA* and *wbaP* could be factors influencing the difference in biofilm formation between K201054 and K201055 and K201060. Expression of capsular polysaccharides is key for *K. pneumoniae* strains to resist complement-mediated killing (25, 26). Mutation *in wcaJ* leads to a deficiency of capsular polysaccharide production and an increase of serum resistance (27), indicating that capsular polysaccharide is not the only factor determining serum resistance. As capsular polysaccharide-deficient strains might resist serum killing by allowing complement deposition and activation (26), this could be a factor contributing to the higher survival rate of strains K201055 and K201060 at the 1 h timepoint in our serum killing assays.

Membrane transporters play a key role in controlling physiological functions such as mediating the flow of molecules between the extracellular environment and the cytosol, including energy sources (28). In our transcriptomic study comparing HMV with

non-HMV isolates, certain dysregulated genes such as *cysU* and *cysP* had been annotated to functions associated to both membrane transport and energy metabolism. Several differentially expressed genes in non-HMV isolates participate in transporter activity, suggesting this pathway to be another component of the complex CPS biosynthesis and HMV regulatory networks, possibly coordinating these features with the intracellular transport of substances or energy metabolism. In addition, we confirmed that the expression of genes involved in carbohydrate metabolism (*pduQ*, *tktA,* and *aceB*) and energy metabolism (*nark*, *metF*, *cysU,* and *cydA*) was also dysregulated. Carbohydrate metabolism and energy metabolism are essential for nutrient acquisition (29) and are increasingly recognized as important contributors to pathogen fitness in host-associated niches (30, 31). CPS biosynthesis and HMV are likely energy-intensive processes, therefore it is intuitive that disruption or repression of genes involved in carbon metabolism may result in reduced CPS biosynthesis and HMV. This supports the notion that the cellular metabolic status in relation to nutrient availability could be a conserved mechanism in CPS synthesis and HMV.

Mutations are not necessarily random but may be the result of a strong genotype-by-environment interaction that enhances recombination function, which is also a manifestation of adaptation effects under appropriate selective pressure (32). In the clinical isolates retrieved in our study, a total of 192 mutations (SNPs and InDels) were identified, the majority of which were located in intergenic regions or were synonymous mutations, with only 12 mutations being non-synonymous. Our genetic studies focused on the coding regions, which might result in the association of some intergenic regions of mutations with the HMV phenotype being excluded. As intergenic mutations might affect the transcriptional activity of genes involved in host interaction, metabolism, and antibiotic susceptibility, identification of mutations in intergenic regions is also important (33). Further studies, particularly focused on upstream regions of genes with unknown functions, will likely uncover novel mechanisms of HMV phenotypic switching.

In summary, our work elucidates that the co-occurrence of HMV and non-HMV phenotypes in the same clonal population may be driven by mutational mechanisms, as well as dysregulation of the expression of genes involved in metabolism and substance transport, with a remarkable impact on the ability to form biofilm.

## MATERIALS AND METHODS

### Case presentation and bacterial strains

On 25 April 2020, a 75-year-old woman was admitted to the Affiliated No.2 People's Hospital in Wuxi, China, for convulsion and unconsciousness. The previous medical records indicated that the patient had underlying conditions (diabetes mellitus and hypertension). Laboratory data revealed a WBC count of 24,000 /mm$^3$, C-reactive protein level of 329 mg/L. A presumed diagnosis of septic shock was made. The inpatient was treated with the empirical administration of cefoperazone sodium and sulbactam sodium. On the next day, two stains (K201054 and K201055) were isolated from blood culture and one strain (K201047) was isolated from cerebrospinal fluid (CSF). Meropenem was administered for the treatment of the infections. During hospitalization, an additional five strains were isolated, two from feces, two from urine, and one from a CSF specimen (Fig. 1). The VITEK two compact System (bioMérieux, France) identified the eight isolates as *K. pneumoniae*. After symptomatic treatment, the inpatient was finally cured and discharged. Bacteria were grown in lysogeny broth (LB) medium and stored at −80°C as glycerol stocks. M9 minimal medium was supplemented with glucose 0.4% to perform mucoviscosity testing. LB medium was used in biofilm formation assays, serum resistance assays and CPS content assays.

## Antimicrobial susceptibility testing

The MIC of antibiotic agents was determined by using the standard Clinical and Laboratory Standards Institute (CLSI) reference broth microdilution method for *K. pneumoniae* clinical strains. *E. coli* ATCC 25922 and *K. pneumoniae* ATCC 700603 were used as the quality control strains (34), and the medium was Müller-Hinton broth medium. The interpretation of the results was based on the 2020 CLSI M100 30th Edition breakpoints, which referred to the EUCAST v.11.0 breakpoints (https://eucast.org/).

## Biofilm assays

Biofilm formation assays were performed as described elsewhere (35, 36). Briefly, bacteria cultures were grown separately until they reached an $OD_{600}$ of 0.2 in LB medium, then adjusting to $1.5 \times 10^7$ CFU/mL. Two-hundred µL of the cell density-adjusted cultures were loaded into the wells of 96-well, flat-bottomed polystyrene plates, and incubated for 20 h at 37°C. At the end of incubation period, planktonic bacteria were removed by washing thrice with 200 µL of distilled water. Wells were dried, treated with 200 µL of 0.5% crystal violet stain for 20 minutes, and washed thrice with distilled water. The bound dye was solubilized with 200 µL of 36% glacial acetic acid and quantified by measuring the optical density at 600 nm ($OD_{600}$). The assay was repeated three times.

## Serum resistance assay

Human serum was collected from healthy volunteers. The bacteria were cultured overnight and then inoculated in LB broth at a ratio of 1:100 to the logarithmic stage. The bacterial suspension was re-suspended to $1.5 \times 10^8$ CFUs/mL. Cell suspensions were thereafter diluted to $1.5 \times 10^6$ CFUs/mL, and 25 µL of $1.5 \times 10^6$ CFU/mL was mixed at a 1:3 (vol/vol) ratio with nonimmune human serum, and incubated at 37°C. A test without human serum was set for each *K. pneumoniae* isolate as the positive control. Heat-inactivated serum was incubated without strains additions as a negative control. Colony counts were determined by the serial dilution method at 1, 2, and 3 h (37). The bacterial survival rate was presented as the percentage of CFU (CFU of the experimental group divided by CFU of the control group).

## CAS agar assays for iron uptake

To elucidate the ability of HMV isolates and HMV isolates to secrete siderophere, we prepared CAS agar plates as described in a previous publication elsewhere (38, 39). Bacterial cell suspensions with grown overnight in LB were adjusted to $1.5 \times 10^8$ CFU/mL and 5 µL were used to inoculate a CAS plate that was cultivated at 37°C for 48 h. The diameters of the orange halos produced on the CAS plates were measured. The assay was repeated three times.

## String test, mucoviscosity testing

The HMV phenotype relies on the classical string test as described (40, 41). The mucoviscosity levels were determined by a natural sedimentation assay. Briefly, isolates were cultivated in M9 minimal medium at 37°C overnight and then adjusted to $1.5 \times 10^9$ CFU/mL, 1 mL of optical density $OD_{600}$-normalized bacteria were precipitated by the action of gravity for 24 h. The absorbance of the supernatant was measured as $OD_{600}$.

## Extraction and quantification of capsule

The capsule was extracted and quantified as described previously (24, 42). Bacteria were inoculated into LB medium and cultured overnight. Five hundred microliters of culture, including LB medium controls, were mixed with 100 µL of 1% Zwittergent 3–14 detergent in 100 mM citric acid, and heated at 50°C for 30 min, mixing every 10 min, The samples were centrifuged at $14,000 \times g$ for 5 min, and 300 µL of the supernatant were

transferred to a new tube and precipitated with ethanol (final concentration 80% vol/vol) at 4°C for 30 min. The precipitate was centrifuged at $14,000 \times g$ for 5 min and the supernatant was discarded, air-dried, dissolved in 500 µL of 100 mM HCl and stored at 4°C overnight (42). Then, 1.25 mL of 12.5 mM sodium tetraborate in concentrated sulfuric acid was added to the samples, mixed, and boiled at 100°C for 5 min. After cooling on ice, 20 µL of 0.5% m-hydroxybiphenyl were added and further incubated for 5 min, 20 µL of 0.5% NaOH was added to the control and the OD was determined at 520 nm. The uronic acids were quantified by comparing the OD of the samples with the glucuronolactone standard curve.

## PFGE profile

The assay was performed as described previously (43). Briefly, overnight bacteria in LB medium were mixed with agarose plugs and treated with lysozyme and proteinase K to lyse the cell wall and precipitate the proteins. The DNA was digested with XbaI (60U) for 4 h before the digested DNA was separated in a 1% agarose gel (6 V/cm, 17.5 h). Bionumerics software (UMPGA clustering method) was used for isolate profiling (44).

## Mouse infection experiments

We used an intraperitoneal infection model to investigate the pathogenicity of the distinct bacterial isolates (four female mice/group). Isolates were prepared by harvesting exponential-phase bacterial cultures in LB broth, washing three times with PBS, then adjusting to $5.5 \times 10^7$ CFU/mL. Five-week-old female BALB/c mice were intraperitoneally injected with 200 ul of working bacterial suspension. The mice were euthanized at 24 h post-inoculation (hpi) and the amounts of bacteria in liver, spleen, and kidney were determined.

## Whole genome sequencing (WGS) and single nucleotide polymorphism (SNP) analysis

All isolates were sequenced using an Illumina HiSeq to generate 150 bp pair-edend reads and 100 x coverage, while K201054 strain was analyzed using PacBio RSII analysis to obtain complete chromosome and plasmid sequences (Majorbio Co., Ltd. Shanghai, China). We used SOAPdenovo v.2. and Unicycler v0.4.6 for *de novo* assembly and obtaining of the complete genome sequence (45). Gene annotations were carried out with the RAST online software (46) (https://rast.theseed.org/FIG/rast.cgi). The MLST profile was assigned according to the online database of MLST v.2.0, while the capsular type of *Klebsiella* strains was determined according to *wzi* gene sequence. Plasmid replicon types were identified by the server Plasmid Finder(47) (https://cge.food. dut.dk/services/Plasmid Finder/). The similarity of the genome sequences was assessed by the JSpeciesWS Online Server(48) (https://jspecies.ribohost.com/jspeciesws). The circular genomic map was generated using CGView(49) (https://paulstothard.github.io/cgview). Virulence determinants were identified using the VFDB server(50) (http://www.mgc.ac.cn/VFs/main.htm). Single-nucleotide polymorphism (SNP) analysis was performed by the kSNP 3.0 based on concatenated genome sequence data of eight *K. pneumoniae* ST412 isolates (51). The mutation sites were annotated by snpEff (https://snpeff.sourceforge.net/SnpEff.html). Single-nucleotide polymorphisms (SNPs) were confirmed by Sanger sequencing.

## Construction of recombinant strains

Construction of recombinant plasmids for gene complementation was performed as described previously (52, 53). The *rmpA* and the native promoter region were amplified by specific primers (Table S3) and cloned into the shuttle vector pBBR1MCS-3 at the KpnI and XbaI sites to generate pBK54. Using the same technique, a second recombinant plasmid containing *wbaP* and its native promoter region was created. After being recovered from *E. coli* DH5α, the recombinant plasmids were independently transformed

into donors (*E. coli S17-λ*). Subsequently, the conjugation assays were carried out using isolates (K201054, K201055, and K201060) as a recipient, and the transconjugants were selected on LB agar plates containing tetracycline (5 µg/mL)(54).

## RNA sequencing and analysis

The eight *K. pneumoniae* were grown overnight in LB medium. The bacterial overnight cultures were diluted at 1:100 in fresh LB medium and grown to mid-exponential phase at 37°C. Bacteria were harvested by centrifugation and total RNAs were extracted using a Spin Column Bacteria Total RNA Purification kit (Sangon Biotech) according to the manufacturer's instructions. RNA-Seq was performed by the Majorbio Bio-pharm Technology Co. Ltd (Shanghai, China) with an Illumina Hi Seq X Ten platform. Gene expression difference analysis was performed using DESeq2. Genes with p-adjust <0.05 or $\log_2 FC > 1$ were considered to be significantly differentially expressed genes (DEGs). Then Gene ontology (GO) functional analysis and genomes (KEGG) pathways functional analysis were performed on differentially expressed genes by cluster profile (55).

## Quantitative reverse-transcription PCR (RT-qPCR)

RT-qPCR reactions were performed as described elsewhere to verify the mRNA levels of differentially expressed genes in the transcriptome (56). The isolates were grown overnight in LB medium. The bacterial overnight cultures were diluted at 1:100 in fresh LB medium and grown to mid-exponential phase at 37°C. Total RNA was harvested from the isolates using the Spin Column Bacteria Total RNA Purification KIT (Sangon Biotech, Shanghai). Then, the extracted RNA was reverse transcribed into cDNA using a TransScript All-in-One First-Strand cDNA Synthesis SuperMix (One-Step gDNA Removal). Finally, the RT-qPCR was performed using a Tip Green qPCR SuperMix (TransGen Biotech Co., Ltd.) in a Mastercycler ep realplex system (Eppendorf, Hamburg, Germany), with an initial incubation at 94°C for 30 s, followed by 40 cycles of 5 s at 94°C and 30 s at 60°C. The internal control gene 16SrRNA was used to normalize the expression of each transcript (see Table S3 in the supplemental material for primer).

## Statistical analyses

Statistical analyses were performed with GraphPad Prism software Version 8. Phenotypic assays were analyzed by *t* test. Statistically significant was defined by $P > 0.05$ (ns), $P < 0.05$ (*), $P < 0.01$ (**), $P < 0.001$ (***), and $P < 0.0001$ (****).

## ACKNOWLEDGMENTS

This research was supported by the National Natural Science Foundation of China (31500114), the Sichuan Province Science and Technology project (2022YFS0631), Project of Luzhou Science and Technology Bureau and Southwest Medical University (2023LZXNYDHZ001), Joint Project of First People's Hospital Of Neijiang and Southwest Medical University (2021NJXNYD06), the Southwest Medical University Foundation (2021ZKZD002), and Top Talent Support Program for young and middle-aged people of Wuxi Health Committee (HB2023021).

## AUTHOR AFFILIATIONS

[1]Department of Pathogenic Biology, School of Basic Medical, Southwest Medical University, Luzhou, China

[2]Department of Laboratory Medicine, Yilong County People's Hospital, Nanchong, China

[3]Department of Microbiology, Tumor and Cell Biology, Karolinska Institutet, Stockholm, Sweden

[4]Department of Clinical Sciences, University of Las Palmas de Gran Canaria, Las Palmas de Gran Canaria, Spain

⁵Department of Dermatology, The Affiliated Hospital,Southwest Medical University, Luzhou, China
⁶Department of Laboratory Medicine, Jiangnan University Medical Center, Wuxi, China
⁷Public Center of Experimental Technology of Pathogen Biology Technology Platform, Southwest Medicine University, Luzhou, China

## AUTHOR ORCIDs

Alberto J. Martín–Rodríguez  http://orcid.org/0000-0003-2422-129X
Renjing Hu  http://orcid.org/0000-0001-6631-1643
Yingshun Zhou  http://orcid.org/0000-0003-4978-0098

## FUNDING

| Funder | Grant(s) | Author(s) |
|---|---|---|
| SPDST \| Sichuan Province Science and Technology Support Program (Science and Technology Project of Sichuan) | 2022YFS0631 | Yingshun Zhou |
| Joint project of southwest Medical University | 2021NJXNYD06 | Yingshun Zhou |
| Southwest Medical University (SWMU) | 2021ZKZD002 | Yingshun Zhou |
| MOST \| National Natural Science Foundation of China (NSFC) | 31500114 | Yingshun Zhou |
| Project of LuZhou Science and Technology Bureau and Southwest Medical University | 2023LZXNYDHZ001 | Yingshun Zhou |
| Top Talent Support Program for young and middle-aged people of Wuxi Health Committee | HB2023021 | Renjing Hu |

## DATA AVAILABILITY

The complete sequences of the isolate in this study, K201054, were deposited in the GenBank database (accession numbers CP110637-CP110639). The draft genome sequences of the other seven isolates were submitted to GenBank under accession numbers JAPJTY000000000, JAPJTX000000000, JAPJTZ000000000, JAPJUA000000000, JAPJUB000000000, JAPJUC000000000 and JAPJUD000000000. Raw sequence reads are available from the NCBI under BioProject accession number PRJNA891137. Raw RNA-seq data are available from the NCBI under BioProject accession number PRJNA891102.

## ETHICS APPROVAL

This study has obtained ethical approval and informed consent of the patient. The ethical code is 2022Y-194. Animal experimental protocols were approved by the Ethics Committee of Experimental Animals, Southwest Medical University. The approval number was 20210225-2.

## ADDITIONAL FILES

The following material is available online.

### Supplemental Material

**Fig. S1 (mSystems00262-24-s0001.tif).** Production of siderophores.
**Fig. S2 (mSystems00262-24-s0002.tif).** PFGE cluster analysis.
**Fig. S3 (mSystems00262-24-s0003.tif).** Circular map of the plasmid.
**Legends (mSystems00262-24-s0004.docx).** Supplemental legends.
**Supplemental Material (mSystems00262-24-s0005.doc).** Raw data for the individual isolates.
**Table S1 (mSystems00262-24-s0006.docx).** Antibiotic susceptibilities of eight isolates.

**Table S2 (mSystems00262-24-s0007.xlsx).** Significant transcriptional changes in GO and KEGG.

**Table S3 (mSystems00262-24-s0008.docx).** Primers.

Open Peer Review

**PEER REVIEW HISTORY (review-history.pdf).** An accounting of the reviewer comments and feedback.

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
