## [Reviewer comments · mSystems]

Co-occurrence of ST412 *Klebsiella pneumoniae* isolates with hypermucoviscous and no-mucoviscous phenotypes in a short-term hospitalized patient

Qinghua Liang, Nan Chen, Wei Wang, Biying Zhang, Jingjing Luo, Ying Zhong, Feiyang Zhang, Zhi Zhang, Alberto J. Martín-Rodríguez, Ying Wang, Li Xiang, Xia Xiong, Renjing Hu, and Yingshun Zhou

Corresponding Author(s): Yingshun Zhou, Southwest Medical University

Review Timeline:

Submission Date:	February 27, 2024
Editorial Decision:	March 25, 2024
Revision Received:	April 22, 2024
Accepted:	May 10, 2024

Editor: Karoline Faust

Reviewer(s): The reviewers have opted to remain anonymous.

Transaction Report:

DOI: <https://doi.org/10.1128/msystems.00262-24>

Re: mSystems00262-24 (Co-occurrence of ST412 *Klebsiella pneumoniae* isolates with hypermucoviscous and no-mucoviscous phenotypes in a short-term hospitalized patient)

Dear Prof. Yingshun Zhou:

The reviewers have now assessed the revised work. Below you will find their comments as well as instructions from the mSystems editorial office. In general, please carefully check English spelling and grammar as reviewers still pointed out problems with English.

Please return the manuscript within 60 days. If you do not wish to modify the manuscript and prefer to submit it to another journal, notify me immediately so that the manuscript may be formally withdrawn from consideration by mSystems.

Revision Guidelines

Sincerely,
Karoline Faust
Editor
mSystems

Reviewer #1 (Comments for the Author):

1. The qRT-PCR data may be easier to interpret if the y-axis is on a log₂ scale with the x-axis crossing at 0 (i.e. decreased transcripts will drop below the x-axis).

2. Line 126 - "non-HMV isolates did not exceed 0.25" is not true as one value is 0.4. May need to revise to state that the mean was 0.25 (range = 0.1-0.4).

3. Figure S4 is compelling and moving it to the main figures should be considered.

4. Lines 283-287 - The authors state that "rmpA is associated with higher biofilm formation ... and wbaP can lead to increased biofilm formation. Therefore, we speculate that this might contribute to the distinct phenotypes exhibited by K201054 as opposed to K201055 and K201060." The referenced phenotypes of rmpA and wbaP on biofilm formation do not logically lead to the authors conclusion. Can the authors please clarify?

In my opinion, I think that what is observed is that K201055 and K201060 are acapsular, which means they are non-mucoid too. However, K201054 is not acapsular, but is non-mucoid (due to lack of RmpD expression), so the phenotypes attributable to capsule abundance persist, but phenotypes (string test) attributable to mucoidy (ie capsule chain length) are absent.

5. Line 300 - Klebsiella uses a Wzy-dependent capsule biosynthesis pathway. It is not the ABC transporter pathway for capsule biosynthesis, which is distinct from the Wzy pathway. PMID: 32680453

6. Please add lines 35-37 of response to reviewers to methods and/or results.

7. Lines 69-70 of response to reviewers - Thank you for including your GO and KEGG analyses. It appears these are located in Table S2, which is not referenced anywhere in the manuscript text. Please reference Table S2 in the text.

Reviewer #2 (Comments for the Author):

This manuscript (mSystems 00262-24) is in its third revision and is continuing to improve. I have a few additional comments for consideration.

- Several references noted were mis-formatted (e.g. 10, 5, 6). I'm seeing this frequently and believe reference programs are to blame. Careful inspection is warranted to make sure all are correct.
- Fig 1 & 2, color scheme for HMV and Non-HMV isolates is reversed. Please be consistent.
- Fig 2, why CFU per 0.1g tissue? As with the sedimentation assay, it is unclear why these authors are choosing to perform and present their data in unconventional ways. Will make it difficult to compare with other studies. [this doesn't really need to be addressed, but something for the authors to consider going forward].
- Fig 3C (serum survival), as with the other panels, please indicate which strains are HMV vs Non-HMV.
- Line 158, the higher levels of colonization by HMV strains at 24 hours is suggestive of increased virulence, but not indicative. Sustained colonization over time or evidence of illness/disease would be needed to for use of the more conclusive term "indicative".
- Gene and protein nomenclature is consistently inaccurate regarding the mutations the authors identified. Genes do not have amino acid substitutions, but instead encode them. Some examples are lines 183, 198, 272, 275.
- Lines 194-196, a mutation in rmpA is described as deletion of a single nucleotide, and an easy assumption is that this led to a premature stop codon. But in Lines 272-273, this mutation is described as R96G (again mixing up gene and protein nomenclature). At the very least, the description of this mutation should be consistent in both locations to avoid confusion.
- Line 216-218, the description of ManB as a "polymerizing signal transduction for envelop biosynthesis" does not make sense with what I understand the function of this enzyme to be, which is the conversion between mannose-1-phosphate and mannose-6-phosphate. This might warrant clarification, particularly as the phrase quoted above does not make sense.
- Line 282, shouldn't these mutations lead to better sedimentation?
- Lines 300-307, the link to ABC-type CPS export is illogical here. Klebsiella capsules are exported by Wzy type complexes. The differences in transporter activity between HMV and non-HMV isolates likely has nothing to do with CPS.
- The HMV strains appear to contain genes encoding additional siderophores (Line 182). It would be appropriate to explain why these strains showed smaller zones of orange in the CAS assay. Are there caveats to this assay that could explain this presumed contradiction?
- In the response to reviewers, there is frequent mention of a modified CPS quantification method. I noted one step that deviated from established protocols (resuspension of the polysaccharide pellet in 100 mM HCl). The authors should explain why this was done, how it improves the protocol, and reference/acknowledge the source if applicable.

Dear Editors and Reviewers,
Thank you for your letter and for the reviewers' comments concerning our
manuscript entitled "mSystems00262-24". The constructive criticism of Editor and
Reviewers has been valuable and very helpful for revising and improving our paper, as
well as for improving the significance of our research. We have studied comments
carefully and have made corrections which we hope meet with approval.
**Responds to the editor:**
**The reviewers have now assessed the revised work. Below you will find their**
**comments as well as instructions from the mSystems editorial office. In general,**
**please carefully check English spelling and grammar as reviewers still pointed out**
**problems with English. Please return the manuscript within 60 days. If you do not**
**wish to modify the manuscript and prefer to submit it to another journal, notify me**
**immediately so that the manuscript may be formally withdrawn from consideration**
**by mSystems.**
Response: Thanks to the editor for the comments. We are very sorry for the mistakes in
this manuscript and the inconvenience caused to the reviewers. We have invited native
speakers to check our English spelling and grammar, and have revised the manuscript
accordingly, so we expect it to meet the journal's standards.
**Reviewer #1 (Comments for the Author):**
**1. The qRT-PCR data may be easier to interpret if the y-axis is on a log₂ scale with**
**the x-axis crossing at 0 (i.e. decreased transcripts will drop below the x-axis).**
Response: Thanks for the reviewer's suggestion, which will greatly improve the quality of
our manuscripts. Based on the reviewer's advice, we have placed the decreased transcripts
below the X-axis in Figure 5E.
**2. Line 126 - "non-HMV isolates did not exceed 0.25" is not true as one value is 0.4.**
**May need to revise to state that the mean was 0.25 (range = 0.1-0.4).**
Response: Thanks to the reviewer for observation, we are sorry for the lack of rigor. We
have corrected the statement from “non-HMV isolates did not exceed 0.25” to “the mean
value of non-HMV isolates did not exceed 0.25” (line 128).
**3. Figure S4 is compelling and moving it to the main figures should be considered.**
Response: We think this is an excellent suggestion. We have moved Figure S4 to the
main manuscript and renamed it Figure 4. In addition, we have revised the sentence
“Introduction of the native *rmpA* and *wbaP* into the corresponding non-HMV strains
restored the HMV phenotype” to “Introduction of native *rmpA* into non-HMV strain
K201054 and *wbaP* into non-HMV strains K201055 and K201060 restored their HMV
phenotypes” (line 204-206).
**4. Lines 283-287 - The authors state that "rmpA is associated with higher biofilm**
**formation ... and wbaP can lead to increased biofilm formation. Therefore, we**
**speculate that this might contribute to the distinct phenotypes exhibited by K201054**
**as opposed to K201055 and K201060." The referenced phenotypes of rmpA and**
**wbaP on biofilm formation do not logically lead to the authors conclusion. Can the**
**authors please clarify?**
**In my opinion, I think that what is observed is that K201055 and K201060 are**
**acapsular, which means they are non-mucoid too. However, K201054 is not**
**acapsular, but is non-mucoid (due to lack of RmpD expression), so the phenotypes**
**attributable to capsule abundance persist, but phenotypes (string test) attributable**
**to mucoidy (ie capsule chain length) are absent.**
Response: Thanks for the reviewer's valuable advice, we apologise for the
error. We looked at the references and found that both *rmpA* and *wbaP* are
associated with biofilm formation. Absence of *rmpA* has been associated
with higher biofilm formation capacity; similarly, mutations in *wbaP* can
lead to increased biofilm formation ability. Therefore, we hypothesize that
the mutations in *rmpA* and *wbaP* could be factors influencing the difference
in biofilm formation between K201054 and K201055 and K201060
(line287-289).
As stated in lines 274-284, we speculate that *rmpA* mutations are
responsible for the non-HMV phenotype of strain K201054 and also result in
better natural sedimentation, which is consistent with evidence that *rmpA* is
associated with the HMV phenotype of *Klebsiella pneumoniae*. The
mutation in *wbaP* resulted in better natural sedimentation and reduced the
overproduction of capsule polysaccharide in the non-HMV strains K201055
and K201060. The reduction in CPS corresponding to a non-HMV
appearance on the plates. This helps explain the differences in natural
sedimentation and capsular polysaccharide of strain K201054 compared to
K201055 and K201060 (line274-284).
**5. Line 300 - Klebsiella uses a Wzy-dependent capsule biosynthesis pathway. It is not**
**the ABC transporter pathway for capsule biosynthesis, which is distinct from the**
**Wzy pathway. PMID: 32680453IF: 10.5 Q1**
Response: Thank you for pointing out this problem in the manuscript. We read this
reference carefully and learned that *Klebsiella* synthesizes capsules via the Wzy-
dependent pathway. We have now made the necessary corrections. In addition, in our
transcriptomic study comparing HMV with non-HMV isolates, certain dysregulated
genes such as *cysU* and *cysP* had been annotated to functions associated to both
membrane transport and energy metabolism. And we have added that to the manuscript
(line299-302).
**6. Please add lines 35-37 of response to reviewers to methods and/or results.**
Response: We gratefully appreciate your valuable suggestion. As suggested by the
reviewers, we have added sample volume and heat-inactivated serum as negative controls
to the serum resistance assay (line376-377).
**7. Lines 69-70 of response to reviewers - Thank you for including your GO and**
**KEGG analyses. It appears these are located in Table S2, which is not referenced**
**anywhere in the manuscript text. Please reference Table S2 in the text.**
Response: Thank you for your observation. In our revised manuscript, we have added
Table S2 which includes GO and KEGG analyses (line238).
**Reviewer #2 (Comments for the Author):**
**This manuscript (mSystems 00262-24) is in its third revision and is continuing to**
**improve. I have a few additional comments for consideration.**
• **Several references noted were mis-formatted (e.g. 10, 5, 6). I'm seeing this**
**frequently and believe reference programs are to blame. Careful inspection is**
**warranted to make sure all are correct.**
Response: Thanks for your careful checks. In our revised manuscript, we have corrected
the format of the references to ensure that the reference format is correct (line520).
• **Fig 1 & 2, color scheme for H MV and Non-H MV isolates is reversed. Please be**
**consistent.**
Response: Thank you for your reminder. We have revised the colour scheme of Figure
2B and 2C to ensure consistency.
• **Fig 2, why CFU per 0.1g tissue? As with the sedimentation assay, it is unclear why**
**these authors are choosing to perform and present their data in unconventional**
**ways. Will make it difficult to compare with other studies. [this doesn't really need**
**to be addressed, but something for the authors to consider going forward].**
Response: We think this is an excellent suggestion. As suggested by the reviewer, we
have converted the data to CFU/g and redrawn Figures 3D, 3E and 3F. In addition, we
have corrected data from “data S7: Raw data for mice infection experiments” in
Supplemental Material 2. Thank you again for your valuable suggestions to improve the
quality of our manuscript.
• **Fig 3C (serum survival), as with the other panels, please indicate which strains are**
**HMV vs Non-HMV.**
Response: Thank you for the above suggestion. In the revised manuscript, we have
redrawn Figure 3C to clearly show the Non-HMV and HMV strains.
• **Line 158, the higher levels of colonization by HMV strains at 24 hours is suggestive**
**of increased virulence, but not indicative. Sustained colonization over time or**
**evidence of illness/disease would be needed to for use of the more conclusive term**
**"indicative".**
Response: Thanks for the reviewer's advice, we apologize for our poor choice of words.
We have rewritten this part as suggested by the reviewer (line160-161).
• **Gene and protein nomenclature is consistently inaccurate regarding the mutations**
**the authors identified. Genes do not have amino acid substitutions, but instead**
**encode them. Some examples are lines 183, 198, 272, 275.**
Response: We sincerely thank the reviewer for their careful reading. We have corrected
the nomenclature in our revised manuscript (line36, 38, 185, 197,200, 274,277).
• **Lines 194-196, a mutation in rmpA is described as deletion of a single nucleotide,**
**and an easy assumption is that this led to a premature stop codon. But in Lines 272-**
**273, this mutation is described as R96G (again mixing up gene and protein**
**nomenclature). At the very least, the description of this mutation should be**
**consistent in both locations to avoid confusion.**
Response: We thank the reviewer for pointing out this issue and we are sorry for the
trouble caused by our incorrect writing. In our revised manuscript, we have revised
“*rmpA*” (R96G) to “*rmpA* (c.285delG)” (line274).
• **Line 216-218, the description of ManB as a "polymerizing signal transduction for**
**envelop biosynthesis" does not make sense with what I understand the function of**
**this enzyme to be, which is the conversion between mannose-1-phosphate and**
**mannose-6-phosphate. This might warrant clarification, particularly as the phrase**
**quoted above does not make sense.**
Response: Thank you for point this out. In the revised manuscript, we have corrected our
statement from “ManB is involved in polymerizing signal transduction for envelope
biosynthesis” to “ManB is distributed at the *cps* locus and might be involved in
converging signal transduction of capsule biosynthesis” (line 218-220).
• **Line 282, shouldn't these mutations lead to better sedimentation?**
Response: Thanks to the reviewer for pointing out this problem. In our revised
manuscript, we have corrected “poor natural sedimentation” to “better natural
sedimentation” (line 274,278).
• **Lines 300-307, the link to ABC-type CPS export is illogical here. Klebsiella**
**capsules are exported by Wzy type complexes. The differences in transporter**
**activity between HMV and non-HMV isolates likely has nothing to do with CPS.**
Response: Thank you for pointing out this problem in the manuscript. We have learned
that Klebsiella synthesizes capsules via the Wzy-dependent pathway rather than ABC-
type CPS export, so we have removed this section in our revised manuscript and we are
sorry again for our mistake (line299).
• **The HMV strains appear to contain genes encoding additional siderophores (Line**
**182). It would be appropriate to explain why these strains showed smaller zones of**
**orange in the CAS assay. Are there caveats to this assay that could explain this**
**presumed contradiction?**
Response: Thanks to the reviewer's careful reading, we are very sorry that our inadequate
expression has caused the reviewer trouble. Manuscript “mSystems00262-24” line 182
refers to genes encoding siderophores on the plasmid pA (*iutA*, *iroBCDN*, *irp2* and *ybts*).
Plasmid pA and plasmid pB were found in eight isolates, both non-HMV and HMV
strains. Based on the reviewer's comment, we carefully checked and found that all the
isolates carried the same genes encoding siderophores and we repeated the siderophore
secretion assay again, the results were still consistent with the original ones.
In addition, we reviewed references related to siderophore and learned that the
exporters were found or suggested to be involved in siderophore release belong to efflux
pumps of the major facilitator superfamily (MFS); the resistance, nodulation, and cell
division (RND) superfamily; and the ATP-binding cassette (ABC) superfamily. There are
two known general mechanisms that lead to iron release from siderophores. The first
comprises the reduction of siderophore-bound Fe(III) to Fe(II) followed by the
spontaneous release or competitive sequestration of the reduced species.
We speculate that siderophore secretion is somehow related to CPS. Siderophore
secretion is a very complex system. Although we have reviewed a lot of literature, we
still cannot explain the reasons for the differences in siderophore secretion between
strains and the details need to be studied.
• **In the response to reviewers, there is frequent mention of a modified CPS**
**quantification method. I noted one step that deviated from established protocols**
**(resuspension of the polysaccharide pellet in 100 mM HCl). The authors should**
**explain why this was done, how it improves the protocol, and reference/acknowledge**
**the source if applicable**
Response: Thanks for the reviewer's reminding, we apologize for the confusion caused by
our inability to clear the expression. The polysaccharide was resuspension by 100mM
HCl, and the content of uronic acid was determined by m-hydroxy-biphenyl method. It is
well known that the principle of the determination of uronic acid content by m-
hydroxybiphenyl method is that the solution of uronic acid and sodium tetraborate sulfate
is hydrolyzed in a boiling water bath, and the hydrolyzed product will further react with
m-hydroxybiphenyl to produce pink derivatives, which have UV absorption, within a
certain range, the UV absorption of the derivative is linear with the content of uronic acid.
The use of 100mM HCl to resuspend the polysaccharide can maintain the acidic PH of
the solution, reduce the interference of the reaction to produce non-specific chromogenic
substances, and bring the detection result of uronic acid closer to the true value. The
modified CPS quantification method we use is based on the original reference ref 24, and
we also add ref 42 (line395, 403).

Re: mSystems00262-24R1 (Co-occurrence of ST412 *Klebsiella pneumoniae* isolates with hypermucoviscous and no-mucoviscous phenotypes in a short-term hospitalized patient)

Dear Prof. Yingshun Zhou:

I am pleased to inform you that your manuscript has been accepted, and I am forwarding it to the ASM production staff for publication. Your paper will first be checked to make sure all elements meet the technical requirements. ASM staff will contact you if anything needs to be revised before copyediting and production can begin. Otherwise, you will be notified when your proofs are ready to be viewed. Please implement the last minor reviewer request in the proofs.

Cover Image Submissions: If you would like to submit a potential Cover Image, please email a file and a short legend to msystems@asmusa.org. Please note that we can only consider images that (i) the authors created or own and (ii) have not been previously published. By submitting, you agree that the image can be used under the same terms as the published article. Image File requirements: TIF/EPS, 7.5 inches wide by 8.25 inches tall (at least 2,250 pixels wide by 2,475 pixels tall), minimum 300 dpi resolution (600 dpi preferred), RGB, and no figure elements, e.g., arrows or panel labels. The legend should be a short description of the image, 1-2 sentences recommended.

Sincerely,
Karoline Faust

Editor
mSystems

Reviewer #2 (Comments for the Author):

In mSystems 00262-24R1, with one exception, the authors have addressed reviewer concerns satisfactorily. Although admittedly a very minor part, I would still argue they are mis-stating the function of manB. The description in this revised manuscript is taken almost directly from the abstract of the referenced article, which, in my opinion, is misleading. I am not an expert on these enzymes and won't insist on this change, but I would encourage the authors to communicate that manB encodes an enzyme that yields one of the sugars that is likely incorporated into CPS-this is the activity that seems more relevant to this study. An interpretation of the current wording is that ManB participates in signal transduction and I'm not sure this is accurate.